# Heritability and complex segregation analysis of naturally-occurring diabetes in Australian Terrier Dogs

**Mei Lun Mui[1], Thomas R. Famula[2], Paula S. Henthorn[1], Rebecka S. Hess[1]\***

**1** Department of Clinical Sciences and Advanced Medicine, School of Veterinary Medicine, University of Pennsylvania, Philadelphia, Pennsylvania, United States of America, **2** Department of Animal Science, University of California–Davis, Davis, California, United States of America

\* rhess@vet.upenn.edu

**Data Availability Statement:** All relevant data are within the manuscript and its Supporting Information files.

## Abstract

The Australian Terrier breed is the breed at highest risk for naturally-occurring diabetes mellitus in the United States, where it is 32 times more likely to develop diabetes compared to mixed breed dogs. However, the heritability and mode of inheritance of spontaneous diabetes in Australian Terriers has not been reported. The aim of this study was therefore to investigate the heritability and mode of inheritance of diabetes in Australian Terriers. A cohort of related Australian Terriers including 383 Australian Terriers without diabetes, 86 Australian Terriers with spontaneous diabetes, and 14 Australian Terriers with an unknown phenotype, was analyzed. A logistic regression model including the effects of sex was formulated to evaluate the heritability of diabetes. The inheritance pattern of spontaneous diabetes in Australian Terriers was investigated by use of complex segregation analysis. Six possible inheritance models were studied, and the Akaike Information Criterion was used to determine the best model for diabetes inheritance in Australian Terriers, among the models deemed biologically feasible. Heritability of diabetes in Australian Terriers was estimated at 0.18 (95% confidence interval 0.0–0.67). There was no significant difference in the effect of males and females on disease outcome. Complex segregation analysis suggested that the mode of diabetes inheritance in Australian Terriers is polygenic, with no evidence for a large effect single gene influencing diabetes. It is concluded that in the population of Australian Terriers bred in the United States, a relatively small degree of genetic variation contributes to spontaneous diabetes. A genetic uniformity for diabetes-susceptible genes within the population of Australian Terriers bred in the Unites States could increase the risk of diabetes in this cohort. These findings hold promise for future genetic studies of canine diabetes focused on this particular breed.

## Introduction

Dogs develop a spontaneous form of diabetes, which shares some characteristics of human type 1 diabetes [1–4]. As in humans with type 1 diabetes, canine diabetes is defined by extreme

**Funding:** Both awards were received by RSH. This work was supported by the American Kennel Club Canine Health Foundation (http://www.akcchf.org, Grant #610); and by a gift from Ms. Catherine Adler. The funders had no role in study design, data collection and analysis, decision to publish, or preparation of the manuscript.

**Competing interests:** The authors have declared that no competing interests exist.

**Abbreviations:** AIC, Akaike Information Criterion; AKC, American Kennel Club.

β-cell deficiency and a requirement for exogenous insulin treatment, without which dogs develop diabetic ketoacidosis [1–4]. Certain pure breed dogs are at increased risk for spontaneous diabetes, suggesting that genetics could be involved in the pathophysiology of the disease [5–9]. The Australian Terrier breed is the breed at highest risk for diabetes in Sweden, Australia, and the United States, where it is 32 times more likely to develop diabetes compared to mixed breed dogs [6, 7, 9]. This is the highest odds of developing diabetes reported in any pure breed dog worldwide. This increased risk of diabetes, observed in three distinct geographic regions, is intriguing because geographic breeding bottlenecks can influence the risk of disease [10]. A recent association analysis of the insulin gene region with diabetes in Australian Terrier and Samoyeds identified an association between a 5.8 Mb region of the *INS* gene and diabetes in these breeds, lending further evidence for a genetic etiology [11].

The heritability and mode of inheritance of canine diabetes has been reported in American Eskimo dogs [12]. However, studies of the heritability and mode of inheritance of diabetes must be breed specific and have not been reported in Australian Terriers. Studies of heritability and mode of inheritance must also focus on specific geographic regions because the unique breeding stock of each region can influence the genetics of disease [10].

The aims of this study were therefore to investigate the heritability and mode of inheritance of diabetes in Australian Terriers in the USA. It was hypothesized that the heritability of diabetes in Australian Terriers is greater than zero and that the mode of inheritance of diabetes in this breed is polygenic. An improved understanding of the genetics of spontaneous canine diabetes could increase the utility of this naturally occurring large animal model and enhance genetic research of diabetes in humans.

## Materials and methods

### Study population

Results of questionnaires completed for a different study were reviewed in detail and data regarding diabetic phenotype, age, and sex of Australian Terriers were retrieved [11]. Data were also collected from an online questionnaire (http://www.vet.upenn.edu/diabetes) that was launched to investigate the prevalence of spontaneous diabetes in dogs across the United States [S1 Appendix: 1, 11]. The survey was promoted by the American Kennel Club (AKC), numerous breed clubs including the Australian Terrier Club of America, 29 academic institutions, 14 private referral practices, and various social media outlets [S1 Appendix: 1, 11]. Data collected for all dogs included age at the time of survey or questionnaire completion, breed, sex, neuter status, presence or absence of diabetes, AKC number if available, names and contact information of owners of immediate dog relatives if known, and owner contact information. Additional data collected for dogs with diabetes included neuter status just prior to diagnosis of diabetes and age at the time of diabetes diagnosis. Australian Terriers recruited from the online survey were included if they were entered into the database by August 31, 2019, but questionnaires were launched as early as 2007. Owners and breeders of Australian Terriers were also contacted directly to collect identical data about other related dogs. AKC pedigrees were used to ascertain pure breed status and ancestry in all study dogs. Australian Terriers with diabetes were included only if an AKC registered pedigree was available for review and if the owner or breeder could be contacted to confirm the diabetic phenotype. Australian Terriers without diabetes or with an unknown phenotype were included only if they were directly related (e.g. sibling, offspring, parent, or grandparent) to a dog with diabetes, as previously described [12]. Australian Terriers bred outside of the United States were excluded. University of Pennsylvania Institutional Animal Care and Use Committee approval for the questionnaire and survey were sought but were waived because dogs were not physically examined. University of Pennsylvania Institutional

Review Board approval was also sought and waived because questions other than contact information pertained to the dog and not to the owner.

## Definition of phenotype

Dogs were defined as cases if their owner or breeder asserted that a veterinarian had diagnosed the dog with insulin treated diabetes. Dogs were defined as controls that do not have diabetes if their owner or breeder reported that the dog had no clinical signs consistent with diabetes (polyuria, polydipsia, polyphagia, or weight loss) and was not being treated with insulin. Dogs were defined as having an unknown phenotype if their owner or breeder could not be contacted. Sex was ultimately classified as female or male because neuter status was unknown in many dogs.

## Estimation of heritability

Heritability was estimated as previously described [12]. The binary disease phenotype (i.e., diabetes and control) dictated the use of logistic regression (logit) to model the risk of disease as a function of explanatory variables (e.g., sex) along with a presumed quantitative genetic contribution. Disease probability was defined as $p_{ij}$ for the $i$-th sex and the $j$-th dog and the logit of this probability was defined as:

$$\theta_{ij} = log[p_{ij}/(1 - p_{ij})].$$

The logit was modeled linearly as a function of sex and a quantitative genotype, as follows:

$$\theta_{ij} = \mu + sex_i + a_j + e_j$$

where $\mu$ is an unknown constant common to all dogs, $sex_i$ is the additive contribution of the $i$-th ($i$ = female or male) sex to the risk of disease, $a_j$ is the additive genetic contribution to the risk of disease for the $j$-th ($j$ = 1,2,3,. . .) dog, and $e_j$ is an unknown random residual contribution to the risk of disease particular to the $j$-th dog. The unknown disease risk ($e_j$) is a function of undetermined environmental factors such as diet, exercise, climate, and veterinary care. Estimation of these unknown effects and predictions of the risk of disease were implemented with the Bayesian statistical package Stan executed with the public domain language R [13, 14]. A hierarchical Bayesian model with weakly informative prior distributions for the unknown effects can help stabilize the estimation process, especially in data sets with a complex pedigree [15].

Defining the prior distributions of these unknown parameters, it was assumed that the intercept ($\mu$) and sex contribution ($sex_i$) were each drawn from a prior distribution of N(0, 16), that the additive genetic effects ($a_j$) were drawn from the multivariate normal distribution N(0, $A\ \sigma_a^2$), with $A$ as the known numerator relationship matrix among all Australian Terriers in the study pedigree and $\sigma_a^2$ defined as the unknown additive genetic variance of disease risk. It was also assumed that $\sigma_a$ was drawn from the positive values of a Cauchy (0, 2.5), as recommended for weakly informative prior distributions [16]. Finally, the residual term ($e_j$), was assumed to be drawn from a standard normal density [i.e., N(0, 1)] as required for this parameterization of a binary trait analysis [17]. Heritability was estimated with $h^2$, which is the proportion of trait variance that is due to additive genetic factors. Accordingly, the heritability of disease risk was estimated as:

$$h^2 = \sigma_a^2/(\sigma_a^2 + 1).$$

The simulation process was conducted across four chains, where each chain was built on a draw of 40,000 total samples, and a "burn-in" process of 15,000 samples followed by thinning to every 25-th sample. In this way, each chain generated 1,000 sample parameter estimates,

and 4,000 samples were generated across the four chains. Convergence of the process was visualized through trace plots of all the unknown values and computation of the Gelman-Rubin statistic for convergence being below 1.05 [18].

## Mode of inheritance

Complex segregation analysis was performed, as previously described, to evaluate whether a single gene of large effect impacts the risk of diabetes in Australian Terriers [12]. Complex segregation analysis was performed using the publicly available package SEGREG, one of several programs available in the S.A.G.E. (v6.4) library [19, 20].

Implementation of this analysis, however, required the elimination of "loops in the pedigree," a well-known challenge in the application of the Elston-Stewart algorithm [21]. Accordingly, prior to the complex segregation analysis, a loop-breaking algorithm was implemented, and dogs were duplicated to remove loops generated by inbreeding [22]. For example, if one dog was related to two different families, this dog was duplicated, and one duplicate remained in one family whereas the other duplicate was assigned to the other family. After implementation of the loop-breaking algorithm, 60 dogs were duplicated and added to a reconfigured pedigree which now included 543 dogs in 45 families. Although likely to decrease the power to detect the linkage of a major locus, the strategy was intended to minimize the impact of this pedigree simplification.

The ensuing analysis applied a model aiming to mimic the logistic regression model outlined above, including a term for the sex of each dog, and a parameter to accommodate shared polygenic terms of family members, as well as the putative major locus effects [23]. Various models with and without Mendelian inheritance were assessed to confirm or exclude the presence of a single gene of large effect [19]. Six models were considered. The simplest was a sporadic model, which assumes no major locus effect, but does consider a term for sex and an accommodation of a polygenic contribution to disease [23]. This was followed by an evaluation of three simple mixed major locus models, considering a dominant, recessive, or codominant major locus all of which follow the expected transmission of alleles outlined by Mendel. That is, for putative major genotypes AA, AB, and BB, the transmission probability for the A allele of these genotypes is set at 1.0, 0.5 and 0.0, respectively. Next, a model that considers environmental transmission of disease, where the polygenic term is removed and the transmission probabilities are set to being identically equal to the estimated allele frequency for all three putative major genotypes, was examined. Finally, a general model was considered where the transmission probabilities of the A allele were estimated from the data set. The Akaike Information Criterion (AIC), in which the goodness-of-fit is evaluated with a penalty for the number of parameters estimated, was used to compare models. Of the biologically feasible models, the one with the smallest AIC was considered the most appropriate for the data.

Age at the time of death or phenotype determination in control dogs, and age at the time of diabetes diagnosis, were normally distributed as determined visually and by the Skewness and Kurtosis tests for normality. Therefore, results for age are reported as mean (and standard deviation) and the two independent samples t-test was used for comparison of these ages. A p value of <0.05 was considered significant. These statistical evaluations were performed using a statistical software package (Stata 14.0 for Mac, Stata Corporation, College Station, TX).

## Results

### Study population

The study population was comprised of 483 Australian Terriers, including 383 dogs unaffected by diabetes (controls), 86 dogs with diabetes (cases), and 14 dogs with an unknown phenotype.

Most dogs (328) were enrolled by directly contacting owners and breeders of Australian Terriers. An additional 128 dogs were enrolled from the survey that was obtained as part of the study investigating the insulin gene region in Australian Terriers, and the remaining 27 dogs were recruited from the online questionnaire (http://www.vet.upenn.edu/diabetes) launched to investigate the prevalence of spontaneous diabetes in dogs across the United States [11, 12]. Contact information for owners and breeders who were contacted directly was made available by these two surveys. According to AKC records, the number of Australian Terrier puppies born from AKC litters between 2007 and 2019 was 5,418 (personnel communication, AKC Canine Health Foundation Raleigh, NC 27675).

The sex and neuter status of the 383 control dogs included 194 females with an unknown neuter status, 171 males with an unknown neuter status, eight neutered females, seven neutered males, two intact females, and one intact male. The sex and neuter status of the 86 diabetic dogs included 33 neutered males, 30 neutered females, 11 males with an unknown neuter status, 8 females with an unknown neuter status, two intact females, and two intact males.

Age of dogs at the time of diabetes diagnosis was known for 74 dogs. Mean (+/- SD) age of these 74 diabetic Australian Terriers at the time of diabetes diagnosis was 9.1 +/- 2.6 years. The mean age at the time of phenotype determination or at the time of death for 49 control dogs in which this age was known, was 11.4 +/- 3.1 years. The age of control dogs at the time of phenotype determination or death was significantly older than the age at which dogs with diabetes were diagnosed with the disease (p< 0.0001).

## Pedigree analysis

A graphic depiction of the pedigree is portrayed in Fig 1. Heritability of diabetes risk in Australian Terrier dogs in a mixed logistic regression model was estimated as 0.18 (95% posterior interval 0.00–0.67). The model included only two sexes (female and male) because the neuter status of many dogs was not known. There was no significant difference in the sex effect on the risk of diabetes detected in the mixed logistic regression model (0.37, 95% posterior interval -0.23–0.94). Age was also not associated with risk of disease.

Complex segregation analysis was performed twice, with and without the sex variable. Results of the complex segregation analysis in the model including sex are reported on a logistic scale in Table 1. Results in Table 1 are reported for females, although males provided similar values. Results of the complex segregation analysis in the model excluding sex are reported on a logistic scale in Table 2. The results of both analyses suggest that the best model to describe the mode of diabetes inheritance in Australian Terriers is the polygenic sporadic model (Tables 1 and 2). In both analyses, the AIC of the sporadic model was smaller than the AIC values of the major locus models (dominant, recessive, and codominant), indicating that the goodness-of-fit of the major locus models was not better than the goodness-of-fit of the sporadic model. In both analyses, the smallest AIC was associated with the general model. However, in both general models the transmission probabilities for the putative A allele estimated from the data were not biologically plausible because they were non- Mendelian and therefore the general models were not considered further.

## Discussion

The fairly small heritability point estimate of 0.18 indicates that in the United States, a relatively small genetic difference between Australian Terriers with and without diabetes, accounts for the difference in phenotypes. Importantly, this low heritability does not indicate that only a small proportion of the risk for diabetes in United States Australian Terriers is genetic. The high risk for diabetes in Australian Terriers, a risk which spans three different continents, speaks to the impactful genetic contribution to diabetes in this breed [6, 7, 9]. The low

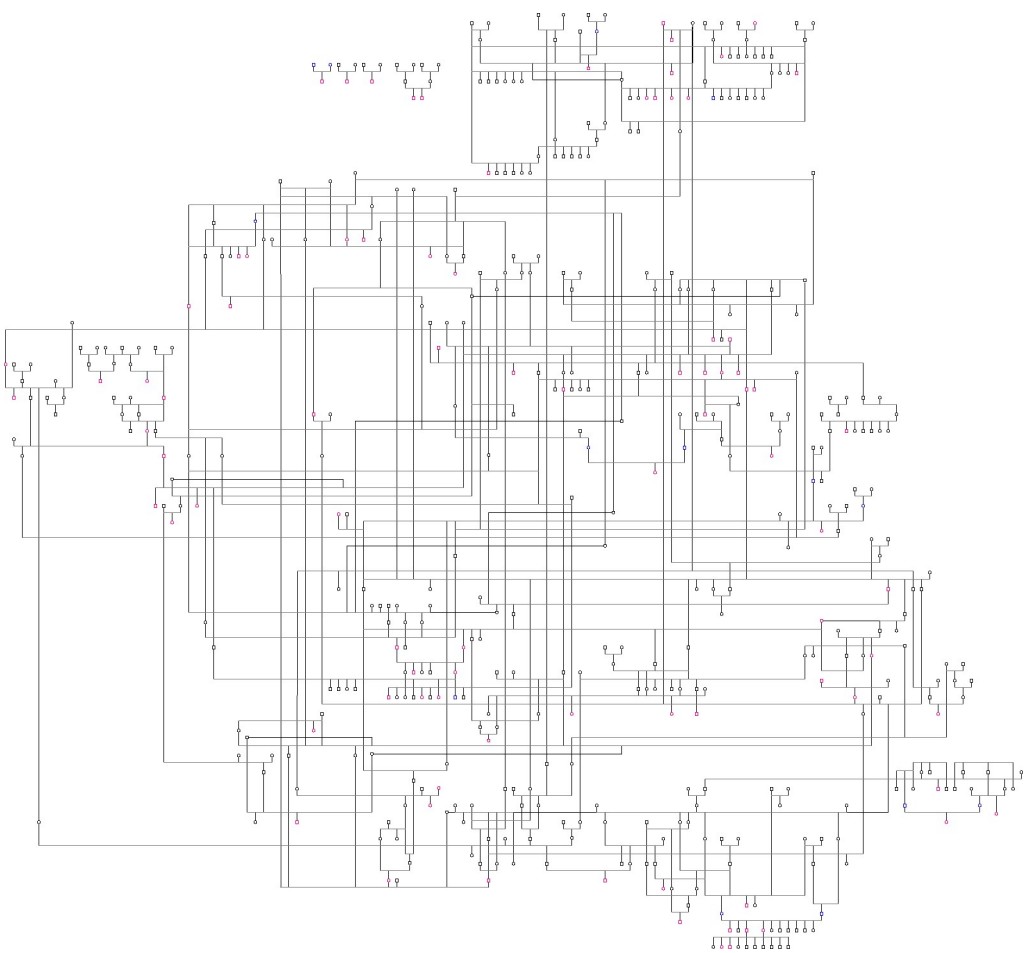

**Fig 1. A graphic depiction of the United States Australian Terrier pedigree under investigation.** Females are represented as circles and males are represented as squares. The illustration includes 86 case dogs with diabetes (designated in red), 383 control dogs without diabetes (designated in black), and 14 dogs with unknown phenotype (designated in blue).

heritability indicates that while genetic contributions to diabetes in Australian Terriers are important, only a small degree of variability in these genes, contributes to the risk of disease. High risk and low heritability of diabetes in Australian Terriers could be explained by

**Table 1. Genetic models tested and their results from the complex segregation analysis of diabetes in Australian Terriers dogs, in a model including sex (results presented for females).**

| Model | q | $\mu_{AA}$ | $\mu_{AB}$ | $\mu_{BB}$ | $\tau_{AA}$ | $\tau_{AB}$ | $\tau_{BB}$ | AIC |
|---|---|---|---|---|---|---|---|---|
| **Sporadic** | – | -1.56 | – | – | – | – | – | 283.5 |
| **Dominant** | 0.17 | 0.41 | -2.38 | -2.38 | 1.0 | 0.5 | 0.0 | 285.0 |
| **Recessive** | 0.83 | -2.38 | 0.41 | 0.41 | 1.0 | 0.5 | 0.0 | 285.0 |
| **Codominant** | 0.39 | -1.56 | -1.56 | -1.56 | 1.0 | 0.5 | 0.0 | 289.5 |
| **Environmental** | 0.48 | -3.32 | -2.57 | 0.41 | = q | = q | = q | 288.5 |
| **General** | 0.01 | -1.83 | 1.07 | -3.65 | 0.75 | 0.13 | 0.99 | 257.4 |

q = frequency of the deleterious allele; $\mu_{AA}$, $\mu_{AB}$, $\mu_{BB}$ are logistic model parameter estimates for the putative major locus genotypes for females; $\tau_{AA}$, $\tau_{AB}$, $\tau_{BB}$ are the transmission probabilities for the putative A allele; AIC is the Akaike Information Criterion.

**Table 2. Genetic models tested and their results from the complex segregation analysis of diabetes in Australian Terriers dogs, in a model excluding sex.**

| Model | q | $\mu_{AA}$ | $\mu_{AB}$ | $\mu_{BB}$ | $\tau_{AA}$ | $\tau_{AB}$ | $\tau_{BB}$ | AIC |
|---|---|---|---|---|---|---|---|---|
| Sporadic | – | -1.54 | – | – | – | – | – | 283.6 |
| Dominant | 0.18 | 0.35 | -2.43 | -2.43 | 1.0 | 0.5 | 0.0 | 285.1 |
| Recessive | 0.82 | -2.43 | 0.35 | 0.35 | 1.0 | 0.5 | 0.0 | 285.1 |
| Codominant | 0.31 | -1.67 | -1.59 | -2.44 | 1.0 | 0.5 | 0.0 | 287.8 |
| Environmental | 0.63 | -2.42 | -3.87 | 2.65 | = q | = q | = q | 284.3 |
| General | 0.89 | -3.14 | -2.31 | 2.39 | 0.74 | 0.09 | 0.19 | 258.5 |

q = frequency of the deleterious allele; $\mu_{AA}$, $\mu_{AB}$, $\mu_{BB}$ are logistic model parameter estimates for the putative major locus genotypes; $\tau_{AA}$, $\tau_{AB}$, $\tau_{BB}$ are the transmission probabilities for the putative A allele; AIC is the Akaike Information Criterion.

inbreeding resulting in genetic homogeneity, a small number of founders, or a small sample size. The wide 95% posterior interval of the estimated heritability indicates that this estimate is imprecise. A larger sample size could increase the accuracy of this estimate. While the study population included only 483 dogs, the total number of AKC registered Australian Terriers born in the United States between 2007–2019 was about 5,400, indicating that the study captured about 9% of the United States Australian Terrier population within this time frame. The low genetic variability contributing to a high risk for disease, coupled with a small overall population all suggest that outbreeding could decrease the risk of diabetes in Australian Terriers.

The only other study to estimate spontaneous diabetes heritability in dogs was performed in American Eskimo dogs [12]. This study estimated heritability of 0.62 for diabetes in American Eskimo dogs and this estimate was also imprecise with a 95% posterior interval of 0.01–0.99.

The heritability of type 1 diabetes in humans is estimated as 0.66–0.88 in different populations [24]. It is possible that the difference in the heritability estimate between the breeds is due to lack of precision in the measurement. It is also possible that the difference in heritability estimates is due to population structure, such as increased genetic homogeneity among Australian Terriers compared to American Eskimo dogs. Finally, it is possible that there truly is more genetic variance contributing to diabetes in American Eskimo dogs compared to Australian Terriers. The relatively low degree of genetic variance influencing the phenotypic expression of diabetes in Australian Terriers, along with the high risk of disease in this breed, make the Australian Terriers an attractive model for future genetic studies of the specific genetic differences impacting diabetes in this breed.

In this study, narrow-sense heritability was calculated because broad-sense heritability cannot be estimated in complex pedigrees such as the one analyzed here. However, narrow-sense heritability accounts for additive genetic variance only. In contrast, broad-sense heritability also considers dominance and epistatic variances. These more complex mechanisms of gene expression (i.e., dominance and particularly epistasis) could also impact the expression of disease, although these terms cannot be evaluated with pedigree information alone [25]. Making use of the available pedigree, all known relationships (including any inbreeding) were incorporated into the estimated heritability. Provided the pedigree in this study is a representative subset of all Australian Terriers, any impact of the presence of this inbreeding on our estimates would be expected to be negligible. The data generated here could also be used to calculate estimated breeding values of diabetes in US Australian Terriers, to help guide future breeding and decrease the risk of diabetes in this breed.

Complex segregation analysis identified polygenic transmission, with several large effect loci, as the most likely mode of diabetes inheritance in United States Australian Terriers.

Similarly, polygenic transmission was identified as the most likely mode of inheritance of diabetes in American Eskimo dogs [12]. Defects in multiple genes are also most commonly associated with inheritance of type I and type II diabetes in humans [26, 27]. Polygenic transmission was identified as the most likely mode of inheritance of diabetes in Australian Terriers because the sporadic model, which considers an accommodation of a polygenic contribution to disease had a lower AIC than all other models except the general model. However, the transmission probabilities for the putative A allele in the general model were not biologically feasible, as they were non-Mendelian, and this model was therefore not considered further.

The research of polygenic complex diseases can be challenging. However, in this particular breed with significant genetic risk and a small degree of genetic variability contributing to disease, overall genetic homogeneity could facilitate genetic discovery. Future genetic research of diabetes in Australian Terriers is therefore promising, and could elucidate the genetics of diabetes in other breeds of dogs. The fact that Australian Terriers on three different continents are the breed at highest risk for diabetes compared to all other breeds of dogs, could suggest that the genetic differences responsible for increased risk of disease occurred before Australian Terriers were dispersed from Australia to other countries. However, specific genetic research in Australian Terriers from different geographic regions is necessary to determine if the genetic architecture of diabetes in this breed is uniform across continents. Sequencing of the 5.8 Mb insulin gene region in Australian Terriers and comparison of this sequence to that found in non-diabetic Australian Terriers holds particular promise for future genetic evaluations because this region has been associated with diabetes in this particular breed [11].

One of this study's limitations is the small sample size. However, about 9% of the United States Australian Terrier population was accounted for in this study. Enrollment in the study was voluntary, and various biases could have been introduced. It is possible that breeders and owners of diabetic dogs were more eager to participate than others, but it is also possible that breeders of dogs with diabetes were hesitant of the exposure, although all data were analyzed anonymously. Therefore, the study population might not be a random sample of the Australian Terrier population at large.

Additional study limitations include recall bias regarding age and neuter status, and the ultimate lack of neuter status data in most dogs. Furthermore, control dogs which were categorized as non-diabetic could have developed diabetes later in life, and could have been misclassified. However, the age of control dogs was significantly older than the age of diabetes onset, minimizing this risk of misclassification. Finally, not all dog owners could be contacted and as a result, some dogs had an unknown phenotype.

In conclusion, the estimated heritability of diabetes in Australian Terriers, the breed at highest risk for diabetes in the United States, Sweden, and Australia is 0.18. This relatively small heritability combined with a high risk of disease could be due to genetic homogeneity, a promising trait for future genetic studies of diabetes in this breed. The mode of inheritance of diabetes in Australian Terriers is polygenic with no evidence of a single gene of large effect impacting the genetic risk of disease. Heritability and other genetic studies of Australian Terriers in other geographic locations are needed to further the understanding of the genetics of diabetes in this breed of dogs.

## Supporting information

**S1 Appendix. Academic veterinary hospitals and private veterinary referral hospitals, which promoted the online survey.**
(DOCX)

## Acknowledgments

The work was carried out at the University of Pennsylvania. The authors declare no conflict of interest and no off-label use of antimicrobials. University of Pennsylvania Institutional Animal Care and Use Committee approval for the questionnaire and survey were sought but were waived because dogs were not physically examined. University of Pennsylvania Institutional Review Board approval was also sought and waived because questions other than contact information pertained to the dog and not to the owner. The authors thank numerous veterinarians, veterinary students and technicians, and dog owners for posting and completing the survey, but especially the American Kennel Club and Australian Terrier Club of America for promoting access to the online survey (http://www.vet.upenn.edu/diabetes).

## Author Contributions

**Conceptualization:** Thomas R. Famula, Rebecka S. Hess.

**Data curation:** Mei Lun Mui, Paula S. Henthorn, Rebecka S. Hess.

**Formal analysis:** Thomas R. Famula, Rebecka S. Hess.

**Funding acquisition:** Rebecka S. Hess.

**Investigation:** Mei Lun Mui, Thomas R. Famula, Paula S. Henthorn, Rebecka S. Hess.

**Methodology:** Mei Lun Mui, Thomas R. Famula, Rebecka S. Hess.

**Project administration:** Rebecka S. Hess.

**Resources:** Rebecka S. Hess.

**Software:** Thomas R. Famula, Rebecka S. Hess.

**Supervision:** Thomas R. Famula, Paula S. Henthorn, Rebecka S. Hess.

**Validation:** Mei Lun Mui, Thomas R. Famula, Rebecka S. Hess.

**Visualization:** Mei Lun Mui, Rebecka S. Hess.

**Writing – original draft:** Mei Lun Mui, Thomas R. Famula, Paula S. Henthorn, Rebecka S. Hess.

**Writing – review & editing:** Mei Lun Mui, Thomas R. Famula, Paula S. Henthorn, Rebecka S. Hess.

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
