## [Decision Letter · Decision Letter 0]

27 Aug 2020

PONE-D-20-19380

Heritability and complex segregation analysis of naturally-occurring diabetes in Australian Terrier Dogs

PLOS ONE

Dear Prof Hess,

Thank you for submitting your manuscript to PLOS ONE. After careful consideration, we feel that it has merit but does not fully meet PLOS ONE’s publication criteria as it currently stands. Therefore, we invite you to submit a revised version of the manuscript that addresses the points raised during the review process.

Many thanks for submitting your manuscript to PLOS One

Two expert reviewers were very impressed with the manuscript and have only made very minor comments for modifications

If you could make the minor modifications suggested, and resubmit your manuscript with a brief response to reviewers that would aid review when resubmitted

I wish to personally thank you for writing such a good paper, and making mine, and the reviewers jobs very easy.

I wish you the best of luck with your revisions

Hope you are keeping safe and well in these difficult times

Thanks

Simon

We look forward to receiving your revised manuscript.

Kind regards,

Simon Clegg, PhD

Academic Editor

PLOS ONE

2. Please amend the manuscript submission data (via Edit Submission) to include author Mei Lun Mui, Thomas R. Famula adn Paula S. Henthorn.

Reviewers' comments:

Reviewer's Responses to Questions

**Comments to the Author**

1. Is the manuscript technically sound, and do the data support the conclusions?

Reviewer #1: Yes

Reviewer #2: Yes

2. Has the statistical analysis been performed appropriately and rigorously? 

Reviewer #1: Yes

Reviewer #2: Yes

3. Have the authors made all data underlying the findings in their manuscript fully available?

Reviewer #1: Yes

Reviewer #2: Yes

4. Is the manuscript presented in an intelligible fashion and written in standard English?

Reviewer #1: Yes

Reviewer #2: Yes

5. Review Comments to the Author

Reviewer #1: This manuscript evaluates the heritability of diabetes in Australian Terriers in the USA. It is well written and clearly defines the populations used and weaknesses within the study (e.g. recall bias and missing data). While the analyses incorporates less than 10% of the breed population, the authors recognise the issues associated with this small percentage of the overall population being used to provide an estimate of heritability.

The authors have performed appropriate analyses of the data and present their findings clearly and concisely. They provide an estimate of heritability of diabetes in Australian Terriers in the USA, across a significant and extended multi-generation pedigree. They do not say whether this pedigree will be used to evaluate the identified genetic susceptibility region in the INS gene – which would be a valuable genetic analyses to perform for the Australian Terriers and may be of relevance to other diabetic susceptible breeds. While not part of this study, further genetic evaluation should be considered in the discussion.

This manuscript will be of interest to breeders and owners of Australian Terriers and highlights the issue of diabetes within the breed.

Reviewer #2: This is an excellent manuscript which examines the heritability of diabetes in Australian Terriers in the USA. It is very very well written, with clear aims and highlights some of the inherent weaknesses within the studies which is nice to see. There is also a decent number of subjects used which is again nice to see. Well written ,well analysed, and very good data collection.

This is a very nice, well researched and very well written article for which the authors should be highly commended. I had great fun reading the article, and would love to see it published very soon. It is of great interest to the dog breeding and diabetes research groups, as well as the wider public. I have made a few very minor comments on the article below. But they are very minor, which highlights the excellent nature of the article. Thank you for making my life as a reviewer easy! The only comment I would say is that it would be great to see some further genetic evaluations and implications discussed but that’s about it really.

Lines 41-44- these sentences could be merged to aid flow

Lines 57-60 are a bit repetitive

Im not sure that table 1 and 2 are adding much as they are just single line tables? These maybe better in text, but I don’t feel strongly and its up to you

Line 284- does this genetic variability with diabetes also link to genetic variability of the dogs? Is there inbreeding co-efficient known for these dogs?

Are there plans to use to genetic susceptibility data to prescreen animals for this disease? That would be a most useful disease diagnostic. Again- not something which needs commenting on, just a thought

It would be nice to see a bit more on further genetic evaluations which maybe useful in the discussion too if you think it would be useful

6. PLOS authors have the option to publish the peer review history of their article (what does this mean?). If published, this will include your full peer review and any attached files.

Reviewer #1: No

Reviewer #2: No

---

## [Author Response · Author response to Decision Letter 0]

3 Sep 2020

The authors thank the Editor and Reviewers for a concise, clear, insightful, and helpful review. All of the Reviewers’ comments have been addressed in the revised manuscript and are also listed below. 

Reviewer 1:

1. They do not say whether this pedigree will be used to evaluate the identified genetic susceptibility region in the INS gene – which would be a valuable genetic analyses to perform for the Australian Terriers and may be of relevance to other diabetic susceptible breeds. While not part of this study, further genetic evaluation should be considered in the discussion.

The authors thank the reviewer for the suggestion. The following sentence has been added on lines 337-340 (in the revised manuscript with trach changes): “Sequencing of the 5.8 Mb insulin gene region in Australian Terriers and comparison of this sequence to that found in non-diabetic Australian Terriers holds particular promise for future genetic evaluations because this region has been associated with diabetes in this breed.” 

Reviewer 2:

1. Lines 41-44- these sentences could be merged to aid flow

The authors thank the reviewer for the suggestion. The sentences on lines 41-43 (in the revised manuscript with trach changes) have been merged, as suggested, and now read as follows: “The Australian Terrier breed is the breed at highest risk for diabetes in Sweden, Australia, and the United States, where it is 32 times more likely to develop diabetes compared to mixed breed dogs [6, 7, 9].”

2. Lines 57-60 are a bit repetitive

Thank you for the suggestion. The following sentence (now on line 64) has been deleted, as suggested: “The heritability and polygenic mode of inheritance of diabetes in Australian Terriers are reported here.”

3. Im not sure that table 1 and 2 are adding much as they are just single line tables? These

maybe better in text, but I donʼt feel strongly and its up to you

Thank you for the suggestions. Table 1 has been deleted and replaced with the following sentence on lines 219-221: “Heritability of diabetes risk in Australian Terrier dogs in a mixed logistic regression model was estimated as 0.18 (95% posterior interval 0.00-0.67).” 

Table 2 was also deleted and its contents were incorporated into the following sentence on lines 222-224: “There was no significant difference in the sex effect on the risk of diabetes detected in the mixed logistic regression model (0.37, 95% posterior interval -0.23-0.94).”

4. Line 284- does this genetic variability with diabetes also link to genetic variability of the

dogs? Is there inbreeding co-efficient known for these dogs?

Thank you for the query. The following sentence has been added to what is now lines 309-313, as suggested: “Making use of the available pedigree, all known relationships (including any inbreeding) were incorporated into the estimated heritability. Provided the pedigree in this study is a representative subset of all Australian Terriers, any impact of the presence of this inbreeding on our estimates would be expected to be negligible.” 

5. Are there plans to use to genetic susceptibility data to prescreen animals for this disease? That would be a most useful disease diagnostic. Again- not something which needs commenting on, just a thought

The following sentence has been added to lines 313-315, as suggested: “The data generated here could also be used to calculate estimated breeding values of diabetes in US Australian Terriers, to help guide future breeding and decrease the risk of diabetes in this breed.” 

6. It would be nice to see a bit more on further genetic evaluations which maybe useful in the

discussion too if you think it would be useful

The following sentence has been added on lines 337-340: “Sequencing of the 5.8 Mb insulin gene region in Australian Terriers and comparison of this sequence to that found in non-diabetic Australian Terriers holds particular promise for future genetic evaluations because this region has been associated with diabetes in this breed.”

---

## [Editor Report · Decision Letter 1]

9 Sep 2020

Heritability and complex segregation analysis of naturally-occurring diabetes in Australian Terrier Dogs

PONE-D-20-19380R1

Dear Prof. Hess

We’re pleased to inform you that your manuscript has been judged scientifically suitable for publication and will be formally accepted for publication once it meets all outstanding technical requirements.

Kind regards,

Simon Clegg, PhD

Academic Editor

PLOS ONE

Additional Editor Comments:

Many thanks for resubmitting your manuscript to PLOS One

As all the comments have been addressed, I have recommended your manuscript for publication

You should hear from the Editorial Office shortly

It was a pleasure working with you, and I wish you all the best for the future

Hope you are keeping safe and well in these difficult times

thanks

Simon

---

## [Editor Report · Acceptance letter]

14 Sep 2020

PONE-D-20-19380R1 

Heritability and complex segregation analysis of naturally-occurring diabetes in Australian Terrier Dogs 

Dear Dr. Hess:

I'm pleased to inform you that your manuscript has been deemed suitable for publication in PLOS ONE. Congratulations! Your manuscript is now with our production department. 

Kind regards, 

on behalf of

Dr. Simon Clegg 

Academic Editor

PLOS ONE